# Influence of Powder Bed Temperature on the Microstructure and Mechanical Properties of Ti-6Al-4V Alloy Fabricated via Laser Powder Bed Fusion

**DOI:** 10.3390/ma14092278

**Published:** 2021-04-28

**Authors:** Lei-Lei Xing, Wen-Jing Zhang, Cong-Cong Zhao, Wen-Qiang Gao, Zhi-Jian Shen, Wei Liu

**Affiliations:** 1School of Materials Science and Engineering, Tsinghua University, Beijing 100084, China; xingll16@mails.tsinghua.edu.cn (L.-L.X.); 13269387926@163.com (W.-Q.G.); shenzhijian@tsinghua.edu.cn (Z.-J.S.); 2Jihua Laboratory, Foshan 528000, China; zhaocc@jihualab.com; 3Arrhenius Laboratory, Department of Materials and Environmental Chemistry, Stockholm University, S-106 91 Stockholm, Sweden

**Keywords:** Ti-6Al-4V alloy, laser powder bed fusion, powder bed temperature, microstructure evolution, mechanical properties

## Abstract

Laser powder bed fusion (LPBF) is being increasingly used in the fabrication of complex-shaped structure parts with high precision. It is easy to form martensitic microstructure in Ti-6Al-4V alloy during manufacturing. Pre-heating the powder bed can enhance the thermal field produced by cyclic laser heating during LPBF, which can tailor the microstructure and further improve the mechanical properties. In the present study, all the Ti-6Al-4V alloy samples manufactured by LPBF at different powder bed temperatures exhibit a near-full densification state, with the densification ratio of above 99.4%. When the powder bed temperature is lower than 400 °C, the specimens are composed of a single α′ martensite. As the temperature elevates to higher than 400 °C, the α and β phase precipitate at the α′ martensite boundaries by the diffusion and redistribution of V element. In addition, the α/α′ lath coarsening is presented with the increasing powder bed temperature. The specimens manufactured at the temperature lower than 400 °C exhibit high strength but bad ductility. Moreover, the ultimate tensile strength and yield strength reduce slightly, whereas the ductility is improved dramatically with the increasing temperature, when it is higher than 400 °C.

## 1. Introduction

Ti-6Al-4V alloy is regarded as an important advanced material and has been widely used in the aerospace industry and biomedical applications because of its low density, high strength, and good corrosion resistance [1,2,3,4,5]. However, their excessive cost and difficulty in processing them severely narrow the expanding application of this alloy. Alternatively, as a near-net forming technique, the laser powder bed fusion (LPBF) can shorten the machining process and improve the mechanical performance; thus, it has attracted extensive attentions from the researchers around the world willing to apply this method to manufacture the Ti-6Al-4V alloy structured parts [6,7,8].

Low-temperature α phase (Hexagonal Close Packed) and high-temperature β phase (Body-Centered Cubic) are two dominating phases in the Ti-6Al-4V alloy. The differences in morphology, size, and volume fraction of the α and β phase always strongly influence the properties of the alloys [9,10]. The metastable acicular martensitic α′ phase with an HCP crystal structure prefers to form during LPBF. This is because the tiny molten pools, formed by the laser melting metal powder during LPBF, undergo rapid solidification (10^3^–10^6^ K/s) [11,12]. The martensitic α′ phase is bad for ductility, whereas the ductility of the lath-shaped α phase and the intergranular β phase, decomposed from the martensitic α′ phase, is improved by heat treatment [13,14]. To improve the ductility of the Ti-6Al-4V products manufactured by LPBF, the post-LPBF heat treatment is often necessary by the martensitic α′ phase decomposing into the α + β phase [1,7,15,16]. However, the post-LPBF heat treatment method has the disadvantage of decreasing the production efficiency.

During printing, the previously deposited layers are always influenced by the thermal effect from the heating, melting, and solidification (latent heat) of the successive layers [17,18,19]. The cumulative heat could lead to the rise in the specimen temperature; however, this increased temperature is still low and hardly causes the α′ martensite to decompose into the α and β phase. In this case, some researchers try to heat the power bed to improve the mechanical properties of the LPBFed structure parts. However, most of the conventional available LPBF systems can only heat the powder bed to less than 200 °C, at which the residual stress generated during LPBF can be reduced, whereas this low temperature hardly induces the α′ martensite to decompose into the α and β phase.

F. Bruckner et al. [20] proposed a model to describe the influence of pre-heating powder bed temperature on the thermometallurgical phenomena in laser cladding initially. B. Vrancken et al. [21] experimentally studied the effect of powder bed temperature on the microstructure evolution of the Ti-6Al-4V alloy, and found that there was little deposition of the α′ martensite when the bed temperature increased to 400 °C. This is the first time that a study reports the α′ martensite decomposition of the Ti-6Al-4V alloy during LPBF. In 2017, Haider Ai et al. [19] investigated the martensitic decomposition and mechanical properties of the Ti-6Al-4V alloy using the powder bed with the temperature of up to 770 °C during LPBF. They found that the α′ martensite could decompose into the α + β phase when the bed temperature increased to 570 °C. However, there are some cavities in the LPBFed Ti-6Al-4V alloy sample, which might damage the mechanical properties. In addition, there is a lack of a systematic study on the microstructural evolution (β phase characteristic) with the increase in in the powder bed temperature during LPBF. Moreover, the relationship between the microstructure and mechanical properties are yet to be further revealed.

Accordingly, in the present study, the Ti-6Al-4V alloy samples were fabricated by LPBF at different powder bed temperatures; subsequently, the microstructure and mechanical properties were detailed, characterized, and measured. All of these were carried out in order to systematically reveal the microstructural evolution with the increasing powder bed temperature during LPBF, in conjunction with the correlation between the microstructure and mechanical properties.

## 2. Experimental Material and Procedures

The Ti-6Al-4V alloy powder produced by plasma rotation electrode process (Grade 1, BDN Technology, China, 15–53 μm) was used. The specific powder composition (in wt.%), measured by an X-ray fluorescence spectrometer (XRF-1500), is shown in Table 1. Nearly all the particles were spherical, and most of the particles had the diameter of no larger than 53 μm, as shown in Figure 1. 

All of the samples were fabricated by the SLM 280 HL facility (SLM Solutions, Lubeck, Germany) with a pre-heating platform. The powder bed can be heated up to 600 °C. The machine is equipped with a fiber laser with a maximum power of 700 W and an operating wavelength of 1075 nm (IPG Photonics Corp, Oxford, MA, USA). The laser beam has a Gaussian profile and a spot diameter of 80 μm. The LPBF processing parameters used in the present study are the laser power (P) of 200 W and the scanning speed (v) of 758 mm/s. The powder layer thickness (t) of 30 μm and a stripe filling strategy with the hatch distance (h) of 110 μm were applied. The laser energy density (E) was 80 J/mm^3^, and this value was obtained using the following equation: $E=\frac{P}{vth}$. The samples were fabricated at unheated, 200 °C, 300 °C, 400 °C, 500 °C, and 600 °C. Multiple-cube specimens with a dimension of 10 mm × 10 mm × 10 mm were obtained. The specific experimental parameters are shown in Table 2. Argon was used to ensure that the oxygen content was below 100 ppm during printing. Three samples were fabricated for each parameter in the present study to reduce the experimental error. And the powder around the fabricated sample was vacuumed between the two processes to reduce the effect of oxidized powder and impurities on the subsequent manufacturing experiments.

The relative density of the LPBFed samples was calculated using the Archimedes method, according to which a body that is immersed in a liquid exhibits an apparent loss in weight equal to the weight of the liquid it displaces. The weight of the samples in ethanol and in the air were tested, respectively. Based on the Archimedes principle, it was as follows: (1)$$m_{\mathrm{air}}{g\;-\;m}_{\mathrm{ethanol}}g=\rho_{\mathrm{ethanol}}\mathrm{gV}$$
where $m_{\mathrm{air}}$ is the weight of the sample tested in the air,$\;m_{\mathrm{ethanol}}$ is the weight tested in ethanol, g is the gravitational constant, and *V* is the volume of the sample, which can be gained by Equation (2), as follows:(2)$$V=\frac{m_{\mathrm{air}}}{\rho_{\mathrm{sample}}}$$
where $\rho_{\mathrm{sample}}$ is the density of the samples, which can be calculated based on Equations (1) and (2).
(3)$$\rho_{\mathrm{sample}}=\frac{m_{\mathrm{air}}\rho_{\mathrm{ethanol}}}{m_{\mathrm{air}}-m_{\mathrm{ethanol}}}$$

And the relative density can be calculated by the dividing of theoretical densities (4.43 g/mm^3^) and measured densities of the Ti-6Al-4V sample.

The morphology and size of the powders and the microstructures of the LPBFed samples were characterized by the scanning electron microscope (SEM, TESCAN) equipped with an electron backscatter diffraction detector (EBSD, Oxford Instruments, Oxford, UK). The samples for SEM and BSE (Backscattered Electron) analysis were prepared by standard mechanical polishing with 400, 800, 1500, and 2000 grit SiC paper, and then polished with alumina solution, with an average diameter of alumina particles of 50 nm. The samples for EBSD analysis were electropolished in a solution of 97% ethanol and 3% perchloric acid. Phase identification was conducted using X-ray diffraction (XRD, D/Max 2500 V) with a Cu target in a conventional X-ray tube, operating at an accelerating voltage of 40 kV and an electron beam current of 30 mA. Further microstructural observation was conducted using transmission electron microscope (TEM, JEOL-2100). The TEM samples were preliminarily prepared by grinding a 3 mm-diameter disk to approximately 30 μm in thickness and then finally thinned by a Gatan-made Precision Ion Polishing System (PIPS) at 5 keV, with a gun angle of ±8°.

Dog-bone shaped specimens with a gauge-section size of 5 mm in length, 2 mm in width, and 1 mm in thickness were cut from the LPBFed samples. Subsequently, all the specimens were mechanically polished to obtain mirror-like surfaces. The cutting position of the tensile sample along with its shape and size are shown in Figure 2. The tensile tests were uniaxially performed at room temperature with an initial strain rate of 10^−3^ s^−1^ using the INSTRON-5969 machine. Each tensile test was repeated for three times to minimize the effect of experimental error on the tensile results.

## 3. Results

### 3.1. Densification Ratio

It is well accepted that the densification ratio plays an important role in determining the ultimate mechanical performance. To eliminate or minimize the effect of fabricated defect on the mechanical properties and to optimize the LPBF parameters, the densification ratios of the LPBFed specimens at different bed temperatures were calculated using the Archimedes method, and the calculated results show that all the LPBFed specimens obtained in the present printing parameters are higher than 99.4%, as shown in Figure 3. Generally, when the densification ratio is higher than 99%, it is recognized as a near-full densification state [22]; thus, in the present study, the effect of the fabricated defect on the mechanical properties can be ignored to some extent.

### 3.2. Phase Characterization

To investigate the phase constitution of the LPBFed Ti-6Al-4V alloy at different powder bed temperatures, the XRD results are shown in Figure 4. The diffraction peaks of the α/α′ phase with HCP structure are found in the sample in the unheated powder bed. It should be noted that it is difficult to differentiate the α and α′ phase by the diffraction peaks, due to their similar crystallographic structure [23]. It is well known that the α′ martensite prefers to form in the Ti-6Al-4V alloy when the high-temperature β phase is rapidly cooled because the atomic diffusion would be inhibited by high cooling rate (over 10^4^ K/s) [11]. As a consequence, the phase transition of β→α′ occurred and the α′ phase formed in the unheated bed [24,25]. The XRD spectrums located in the left were obtained between 32 and 65°, with a step size of 0.02° and a counting time of 1 s/step, which is a coarse scanning strategy and displays a severe signal-to-noise ratio. The β phase is difficult to be detected in this speed. However, the XRD spectrums located in the right were obtained between 55° and 60°, with a step size of 0.01° and a counting time of 5 s/step, which is a fine scanning strategy and displays a slight signal-to-noise ratio. Moreover, the β phase could be well detected under this condition. When the powder bed temperature increases to 200 °C and 300 °C, there is still only α/α′ phase and no β phase detected. However, as the powder bed temperature increases to 400 °C, the diffraction peaks of the β phase appear, indicating that the α′ martensite decomposed into the α and β phase. More β phase is found at the powder bed temperatures of 500 °C and 600 °C. It should be noted that the diffraction peaks cannot be found when the phase content is less than 5%, and thus, it is not sufficient to reveal the phase decomposition only by XRD; more characterization work should be performed.

### 3.3. Microstructure Characterization

A typical microstructure of the Ti-6Al-4V alloy sample manufactured using an unheated powder bed is given in Figure 5a, exhibiting martensitic α′ laths, which is consistent with the XRD results (Figure 4), which does not exist in the β phase. The martensitic α′ appears not to be homogeneous, and the thickness of the α′ lath exhibits a large difference. The microstructure at the powder bed temperature of 200 °C (Figure 5b) is almost similar to the unheated bed sample. However, as the powder bed temperature increases to 300 °C, compared with the microstructure at 200 °C, the large difference in the morphology takes place, even though there is still no β phase observed, as shown in Figure 5c. When the powder bed temperature increases to 400 °C, the β phase (the white area) appears, as highlighted by the red arrow in Figure 5d. Figure 5e shows the microstructure at the powder bed temperature of 500 °C, and more β phase is observed along the boundaries of the α phase laths, which is highlighted by the blue arrow. The increased powder bed temperature can enhance the decomposing martensitic α′ laths and the coarsening α lath. Thus, much thicker α phase laths and more β phase (highlighted by the purple arrows) are found at the powder bed temperature of 600 °C, as shown in Figure 5f.

For further understanding the phase transformation during LPBF, TEM was performed to observe the microstructures at different bed temperatures from a more microscopic perspective, as shown in Figure 6. It can be seen from Figure 6a that the microstructure exhibits lath-shaped morphology with HCP crystal structure, which can be attested by the corresponding selected area electron diffraction patterns (SADP) displayed at the bottom right corner of Figure 6a. Moreover, the high-density dislocations existing inside the laths reveal that a large amount of strain and lattice defects were generated during LPBF. These laths are recognized as α′ martensite, which is consistent with the previous results reported by other researchers [1,2,3]. During LPBF, the rapid cooling leads to the formation of metastable martensitic with an HCP crystal structure, which is bad for ductility [26]. To solve this problem, the preheated powder bed works as an option [19]. When the powder bed temperatures of 200 °C and 300 °C were utilized, the microstructure after manufacturing still appeared as martensitic α′ laths, but the amount of the dislocations inside α′ laths decreased, as shown in Figure 6b,c. As the powder bed temperature increased to 400 °C, the boundary became clear, and a thin layer between the α laths formed; this thin layer can be proved as a β phase by the SADP displayed at the bottom left corner in Figure 6d. This is consistent with the SEM result shown in Figure 5. The width of the α laths increases with the powder bed temperature increasing to 500 °C and 600 °C, as shown in Figure 6e,f. The thickness of the β phase layer is approximately 5–10 nm, as highlighted by the blue arrows in Figure 6d–f. Scan transmission electron microscope (STEM) was performed to reveal the element content in the β phase layer distributing along the boundaries at the powder bed temperatures of 400 °C, 500 °C, and 600 °C, and the results are shown in Table 3. It can be found that the V elements enrich in the β phase layer, and this phenomenon is more obvious with the increasing powder bed temperature. By the same token, it is reasonable to conclude by the TEM results that the martensitic α′ laths decompose into α laths and the intragranular β phase by the V element partitioning, when the powder bed temperature is equal to and above 400 °C; meanwhile, the α laths coarsen with the increasing powder bed temperature. 

To reveal the orientation relationship of the α and β phase, the high resolution TEM (HRTEM) image was obtained, as shown in Figure 7. Figure 7b presents the Inverse Fast Fourier Transition (IFFT) image of the boundaries between the α and β phase. The (1(—)100) atomic row spacing in the α grain is 0.255 nm, and the (011(—)) atomic row spacing in the β grain is 0.299 nm. Figure 7c reveals the FFT image corresponding to Figure 7b. It can be concluded that the β grain displays a conventional orientation relationship (OR) of [011]_β_//[0001]_α_ in the α grain.

To further identify the characteristics of α/α′ laths in statistics, the EBSD maps of Ti-6Al-4V alloy after LPBF at different powder bed temperatures are shown in Figure 8. It is obvious that these microstructures consist of a large amount of long and straight laths. As described above, these laths have been confirmed as α/α′ phase with an HCP structure. It should be noted that the intragranular β phase with the BCC structure is difficult to identify due to its thin thickness (Figure 6). Here, to quantitatively analyze the lath-shaped microstructure evolution with the increasing powder bed temperature, the aspect ratio (it is defined as the ratio of length to width of the lath) was calculated by analyzing the EBSD data. It is found that the average aspect ratios of laths at different powder bed temperatures are 5.06, 4.77, 4.46, 4.05, 3.67, and 2.75, respectively, and they decrease with the increasing powder bed temperature, as shown in Table 4.

### 3.4. Mechanical Properties

The room-temperature tensile results of the Ti-6Al-4V alloy after LPBF at different powder bed temperatures are shown in Figure 9. The specimens fabricated in unheated powder bed show the highest strength and the worst ductility. As the powder bed temperature increases, the tensile performance exhibits a trade-off phenomenon in strength and ductility; specifically, the ultimate tensile and yield strengths decrease, whereas ductility improves. The ultimate tensile and yield strengths decreased by 6.7%and 1.5%, respectively, and the elongation increased by 66.0%, when the powder bed was heated to 600 °C. It should be noted that, although the strengths (especially the ultimate tensile strength) decrease, they do not decrease noticeably and still maintain at a high level with the increasing powder bed temperature. By contrast, ductility apparently improves as the powder bed temperature increases, especially when the temperature is above 400 °C, as shown in Figure 9c. In addition, in all cases, the elastic modulus remains unchanged, with a value of around 1168 GPa. Based on what is described above, the negative mechanical properties can be eliminated by heating the powder bed, and it is reasonable to conclude that, in the present study, the excellent comprehensive mechanical properties are obtained at the powder bed temperature of 600 °C.

## 4. Discussion

### 4.1. Microstructure Evolution with the Increasing Powder Bed Temperature during LPBF

In the present study, it is clear that the powder bed temperature plays an important role in determining the microstructure after LPBF. The ultimate microstructure is dominated by the α′ martensite, with a large aspect ratio of 5.06, in the un-preheated powder bed. It is also well known that the structure parts are built layer by layer during LPBF, and it is reported that the cooling rate during LPBF is approximately 10^3^–10^8^ K/s [11,12], which is much higher than the 1500 K/s detected during water quenching [27]. This high cooling rate during LPBF suppresses the atomic diffusion and the phase transformation of β→α (occurring in the equilibrium state) is inhibited; alternatively, the β→α′ occurs during LPBE. 

When the powder temperature is elevated to 200 °C, the sample temperature during manufacturing is still low, which means that the V element is hardly diffused [18,28,29,30], which is regarded as the indicator of the occurrence of martensite decomposition. However, the increasing powder bed temperature widens the laths slightly, which leads to a decrease in the aspect ratio (4.77), lower than 5.06 obtained in the un-preheated condition [17,31,32]. 

As the powder bed temperature increases up to 300 °C, the boundaries between the martensitic α′ laths become clear. However, the specimen temperature during printing is still lower than the martensitic decomposition temperature of ~550 °C for the Ti-6Al-4V alloy [33]. In addition, the higher temperature decreases the aspect ratio due to the widening of the α′ lath.

During LPBF, the previous deposited layers are affected by the thermal-cycle effect from the laser heating as well as the melting and solidification latent heat of the subsequent successive layers. As the powder bed temperature increases to 400 °C, though it is lower than the martensitic decomposition temperature of ~550 °C for the Ti-6Al-4V alloy, the cumulative heat generated due to the thermal-cycle history during LPBF could lead to the specimen temperature rising above 550 °C [17,18,19,34]. It is difficult to directly detect the temperature evolution process of the specimens during LPBF. Xu et al. [34] drew the temperature evolution schematic diagrams of the Ti-6Al-4V alloy during LPBF and thought the temperature of the sample gradually increases at the initial stage, and then decreases to the powder bed temperature. Based on this observation, it can be concluded that the sample during LPBF can be heated, while the peak temperature must be higher than the pre-heated temperature of powder bed. Consequently, the peak temperature of the sample during LPBF could be above 550 °C, though the powder bed temperature is only 400 °C. In this case, the diffusion of the V element in α′ laths was motivated and the α phase initiated to nucleate along the α′ boundaries, during which and the subsequent process, the V atoms was expulsed from the newly formed α phase and preferred to diffuse towards the lath boundaries [2,18]. The enrichment of the V element caused the formation of the β phase. Ultimately, the martensitic α′ phase decomposed into the α + β phase. However, the decomposition of the α′ phase is not fully completed and the amount of the β phase is small at the powder bed temperature of 400 °C, which is attributed to the short-residence time in the decomposition temperature range [34]. In addition, the aspect ratio of laths decreases due to the widening of the α′ lath at the powder bed temperature of 400 °C.

When the powder bed temperature is elevated to 500 °C and 600 °C, more energy is supplied, which tells us that the peak temperature of the sample during LPBF is much higher than the martensitic decomposition temperature of ~550 °C. It also shows that the cumulative residence time of the previous deposited layer in the α′ martensitic decomposition temperature range is much longer, resulting in more β phase forming and the widening of the laths [17,31].

### 4.2. Influence of Microstructure on Mechanical Properties

It is well known that the microstructure determines the mechanical properties. In the present study, it is evident that the mechanical properties are strictly related to the microstructure of the LPBFed Ti-6Al-4V alloy, at different powder bed temperatures. 

The thickness of the martensitic α′ laths after LPBF in the unheated powder bed is shown to be relatively fine and un-homogeneous, which makes it easy to prevent the dislocation slip, leading to the crack initiation and propagation at the weak location in the microstructure during deformation [19,35]. In addition, there exist high-density dislocations induced by the high residual stress in the α′ phase, as shown in Figure 5a, which could generate stress accumulation. As a result, the microstructure obtained in the unheated powder bed shows the highest strength but the worst ductility. 

When the powder bed temperature is below 400 °C, the widening and decomposition of the α′ martensite laths are limited, which should be responsible for the slight decrease in strength and the improvement of ductility.

As the powder bed temperature increases to 500 °C and 600 °C, the microstructure becomes coarsening and the β phase increases. The widening laths would increase the dislocation slip length and decrease the stress accumulation during deformation [36]. Moreover, the slip resistance between the α and β phase is lower than that between the α′ laths [9,10,35,37]. As a result, ductility is improved, but the strength decreases with the increasing powder bed temperature [37,38]. It should be noted that the ductility of the samples manufactured at the powder bed temperatures of 500 °C and 600 °C is superior to that reported by other researchers using the same method of heating the powder bed [19]. This may be due to the fact that the densification of the present manufactured samples is better than that in ref. [19]. In a word, increasing the powder bed temperature enhances the comprehensive mechanical properties due to the β phase’s precipitating and the α lath’s widening, which is better suited for the various applications of the LPBFed Ti-6Al-4V alloy.

## 5. Conclusions

In the present study, Ti-6Al-4V alloy samples were fabricated by laser powder bed fusion, at different powder bed temperatures. The microstructure and the corresponding mechanical properties of the LPBFed Ti-6Al-4V alloy were investigated. The main conclusions are as follows: 

All of the Ti-6Al-4V alloy samples fabricated by laser powder bed fusion at different powder bed temperatures exhibit a near-full densification state with the densification ratio of above 99.4%. 

When the powder bed is unheated or the heating temperature is lower than 400 °C, the microstructure is dominated by the α′ martensite with a large aspect ratio. When the increasing temperature is equal to or above 400 °C, the diffusion and redistribution of the β-phase stabilizer of the V element in the α′ laths matrix occurs, resulting in the enrichment of V atoms along the grain boundaries. Thus, ultimately, the α′ lath decomposes into a lath-shaped α and an intergranular β phase. In addition, with the increase in the powder bed temperature, the α/α′ laths widen, and the aspect ratio decreases. 

When the powder bed is unheated or the heating temperature is lower than 400 °C, the specimens exhibit high strength but bad ductility because of the existence of the α′ martensite. As the temperature elevates to a value higher than 400 °C, the ultimate tensile strength and yield strength reduce slightly, whereas the ductility is dramatically improved with the increasing powder bed temperature, which is attributed to the α′ martensite decomposing into the α and β phase and the widening of α laths. 

The powder bed temperature has a significant influence on microstructure evolution and mechanical properties of the Ti-6Al-4V alloy during LPBF. The excellent combination of mechanical properties, including strength and ductility, appears at the powder bed temperature of 600 °C.

## Figures and Tables

**Figure 1 materials-14-02278-f001:**
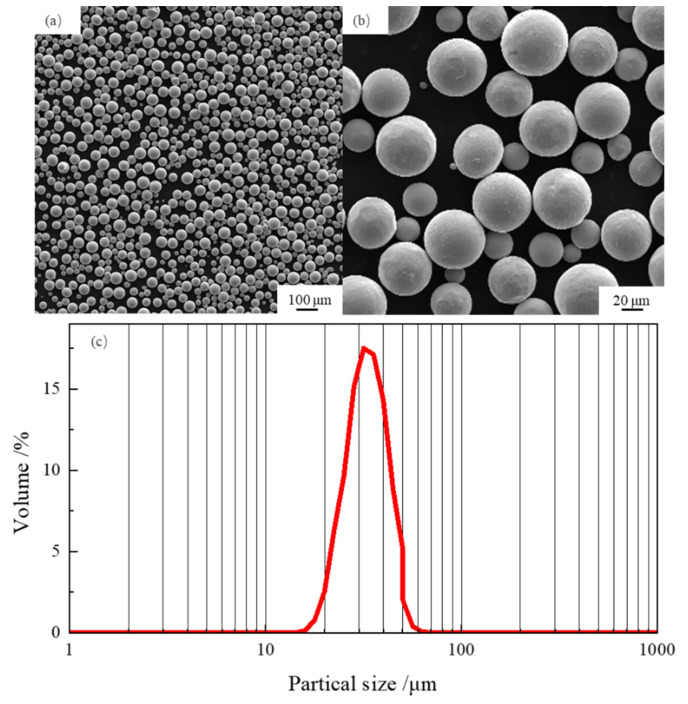
(**a**,**b**) morphology; (**c**) partial size distribution of the Ti-6Al-4V alloy powder.

**Figure 2 materials-14-02278-f002:**
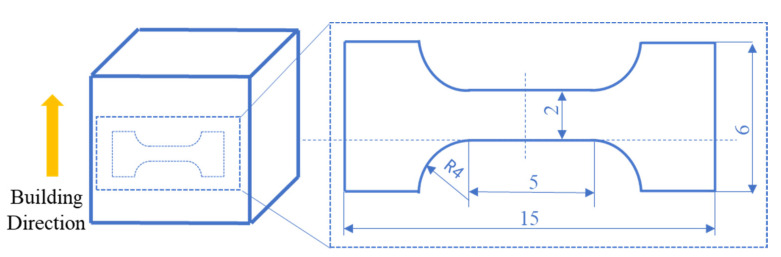
The cutting position with the shape and size of the tensile specimen for mechanical tests (dimensions of the sample given in mm).

**Figure 3 materials-14-02278-f003:**
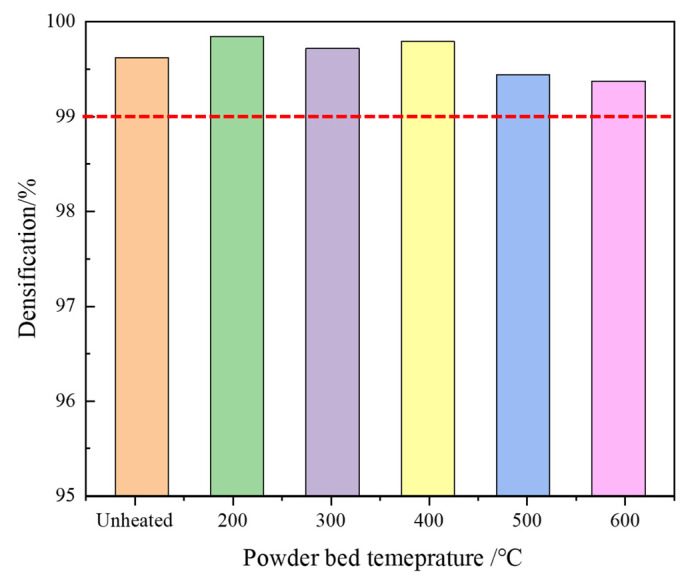
The densification ratio of the Ti-6Al-4V alloy samples manufactured at different powder bed temperatures.

**Figure 4 materials-14-02278-f004:**
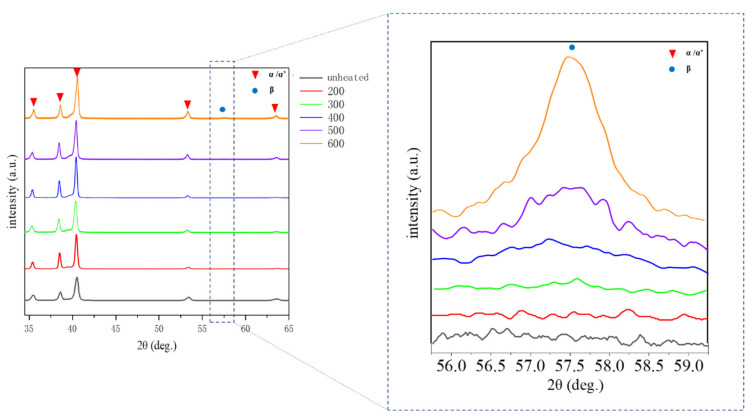
The XRD patterns of the Ti-6Al-4V alloy samples manufactured at different powder bed temperatures.

**Figure 5 materials-14-02278-f005:**
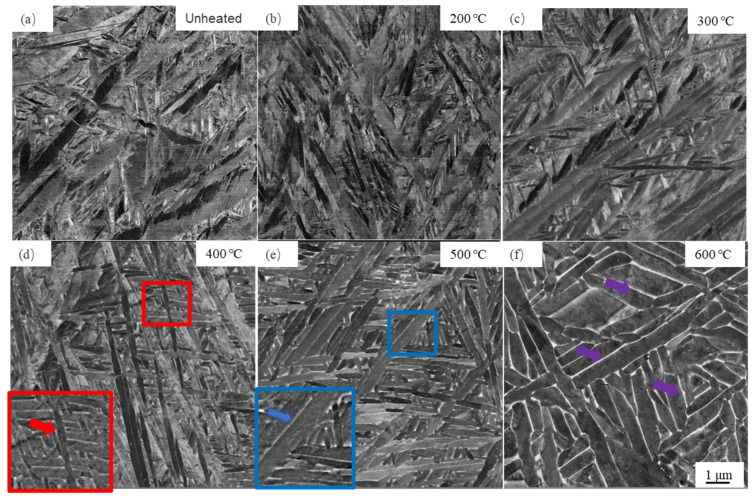
BSE micrographs of the Ti-6Al-4V alloy produced by LPBF in the case of (**a**) unheated, (**b**) 200 °C, (**c**) 300 °C, (**d**) 400 °C, (**e**) 500 °C, and (**f**) 600 °C.

**Figure 6 materials-14-02278-f006:**
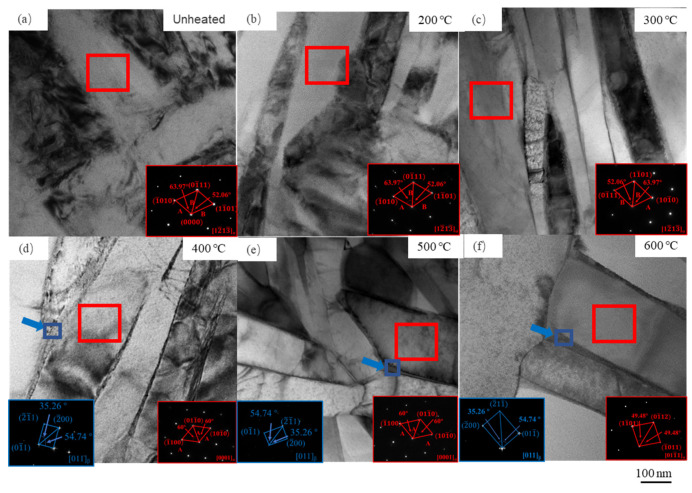
Bright field images and the corresponding selected area electron diffraction of the Ti-6Al-4V alloy after LPBF in the case of (**a**) unheated, (**b**) 200 °C, (**c**) 300 °C, (**d**) 400 °C, (**e**) 500 °C, and (**f**) 600 °C.

**Figure 7 materials-14-02278-f007:**
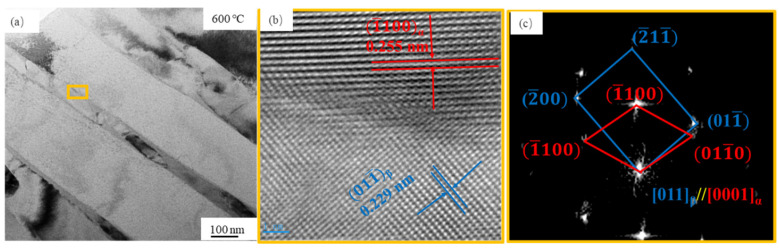
(**a**) Bright field image, (**b**) HRTEM IFFT image, and (**c**) FFT image of the Ti-6Al-4V alloy after LPBF at the powder bed temperature of 600 °C.

**Figure 8 materials-14-02278-f008:**
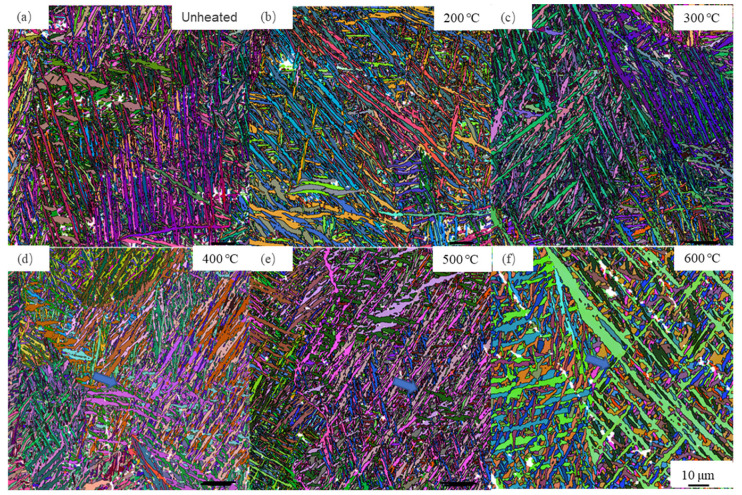
EBSD maps of the Ti-6Al-4V alloy after LPBF in the case of (**a**) unheated, (**b**) 200 °C, (**c**) 300 °C, (**d**) 400 °C, (**e**) 500 °C, and (**f**) 600 °C.

**Figure 9 materials-14-02278-f009:**
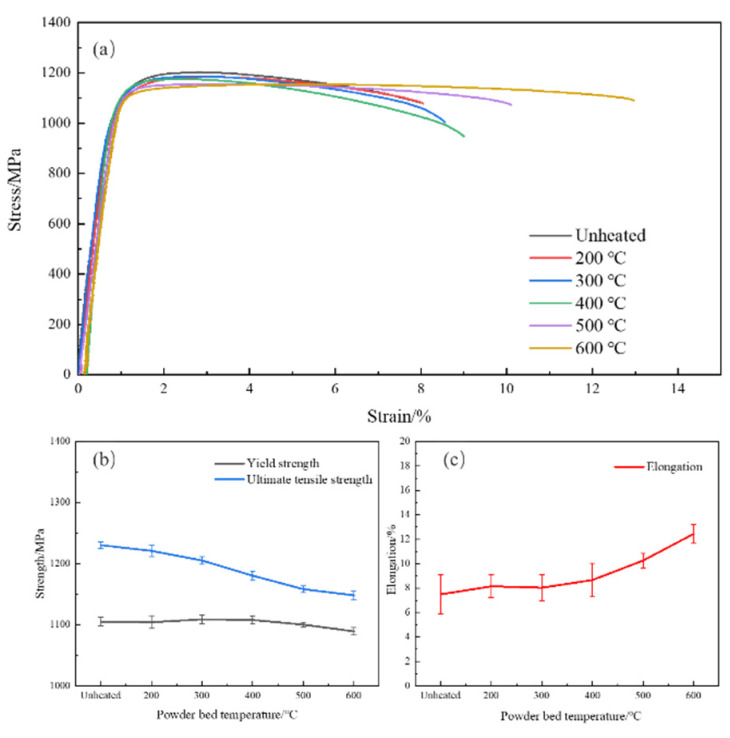
(**a**) Engineering stress-strain curves, (**b**) strengths, and (**c**) elongation of the Ti-6Al-4V alloy after LPBF at different powder bed temperatures.

**Table 1 materials-14-02278-t001:** Powder elemental composition (weight %).

Elements.	Al	V	Fe	Na	Ti
Contents/wt.%	6.14	3.82	0.183	0.135	balanced

**Table 2 materials-14-02278-t002:** Laser powder bed melting fabricated parameters

Bed Temperature	Laser Power/W	Scanning Speed/mm/s	Powder Layer Thickness/μm	Hatch Distance/μm	Energy Density/J/mm^3^
Unheated	200	758	30	110	80
200 °C					
300 °C					
400 °C					
500 °C					
600 °C					

**Table 3 materials-14-02278-t003:** Element content in the intragranular β phase of Figure 5d–f.

Element Content, wt.%	400 °C	500 °C	600 °C
Ti	87.65	83.80	82.32
Al	3.43	3.28	4.65
V	8.92	12.91	13.03

**Table 4 materials-14-02278-t004:** The aspect ratios at different powder bed temperatures.

Grain Statistics	Unheated	200 °C	300 °C	400 °C	500 °C	600 °C
Aspect ratio	5.06	4.77	4.46	4.05	3.67	2.75

## Data Availability

Not applicable.

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
