# Peer review of "Influence of Powder Bed Temperature on the Microstructure and Mechanical Properties of Ti-6Al-4V Alloy Fabricated via Laser Powder Bed Fusion"

_materials, 2021, doi:10.3390/ma14092278_

Round 1

Reviewer 1 Report

The article is devoted to a topical topic.
It is of scientific and practical interest to readers.
However, the presentation of the material needs some work.
1) The photograph in Figure 1 is poorly scaled. Powder particles are not spherical in the photo. It would be possible to add a ruler with dimensions for a few powder particles. This will allow you to better estimate their size.

2) In 2. Experimental material and Procedures, you need to add a description of the Archimedes method. Specify how exactly the density was determined. Why does bed temperature affect density? What is the mechanism of the process? 

3) To visualize the number and shape of pores, you can give a photo of un-etched samples.

4) Figure 8 is of poor quality. The lines are hard to see. I recommend building it on a different scale. The Y-axis can be specified not from "0", but from "1000" to "1300". This will push the lines of the stretch charts apart. In the discussion and conclusions, it is necessary to quantitatively indicate how much the strength decreased and the plasticity increased when the substrate was heated to 600 °С.

5) The work does not indicate in which direction, according to the sample growth scheme, samples for mechanical tests were cut. This has a significant effect on the values ​​of the mechanical properties. I recommend looking at the following work on this issue https://doi.org/10.3390/machines8040079.

6) The photo in Figure 4 can be improved.

7) The article has a large number of design flaws that need to be eliminated.

Author Response

Dear Editors and Reviewers:

Thank you for your reviewing and giving your kind and meaningful comments concerning our manuscript entitled “Influence of powder bed temperature on the microstructure and mechanical properties of Ti-6Al-4V alloy fabricated via la-ser powder bed fusion” (ID: materials-1191348). The comments are all valuable and very helpful for revising and improving our paper, as well as the important guiding significance to our researches. We have studied the comments carefully and have made some corrections which we hope meet with approval. Revised section is clearly highlighted using the "Track Changes" function in Microsoft Word. The main corrections in the paper and the responds to the reviewer’s comments are as flowing:

Responds to the reviewer’s comments:

  1. Response to comment: (The photograph in Figure 1 is poorly scaled. Powder particles are not spherical in the photo. It would be possible to add a ruler with dimensions for a few powder particles. This will allow you to better estimate their size.)

Response: It’s true as Reviewer suggested. To present the powder size, we tested the powder size distribution by Mastersizer 2000 laser particle size analyzer, as shown in Fig.1(c).

  1. Response to comment:( In 2. Experimental material and Procedures, you need to add a description of the Archimedes method. Specify how exactly the density was determined. Why does bed temperature affect density? What is the mechanism of the process?)

Response: As Reviewer suggested that we have added the description of the Archimedes method and the exact calculating process of the density.

It is true as Reviewer suggested that the mechanism that the effect of the powder  bed temperature on the density is very important. However, the physical process of LPBF is complicated: the beam energy is absorbed by powder bed and the powder particles is heated, and then a melt pool forms accompanying with a sudden release the surface energy, wetting of melt pool within the surrounding stochastic powder particles and former coalesced layer, the thermocapillary convection in the melt pool induced by temperature gradients and surface tension gradients, the weld pool oscillation under the pulsed laser recoil force and metallic vapor, the shear stress in gas–melt interface from the flowing shield gas. The physical process, as well as the manufacturing defects and material density are severely affected by the powder bed temperature. It needs more experiment and simulation evidence to clarify the mechanism. In the present study, a large number of experimental observations have confirmed that there is little defect and few pores are found in all the fabricated samples. And the density would be investigated detailedly in our future work.

  1. Response to comment:( To visualize the number and shape of pores, you can give a photo of un-etched samples.)

Response: We are very sorry for the negligence of the preparation description for the SEM and BSE samples. They were prepared by standard mechanical polishing with 400, 800, 1500 and 2000 grit SiC paper, and then polished in alumina solution with an average diameter of alumina particles of 3 μm and 50 nm. This statement has been added in the paper. Thus, they are un-etched samples. It is true as Reviewer suggested that the number, shape and distribution of pores are very important. However, in the present study, many experiments observations have confirmed that and few pores are found in all the fabricated samples. And the slight pores would be investigated detailedly such as using Micro-CT in our future work.

  1. Response to comment:( Figure 8 is of poor quality. The lines are hard to see. I recommend building it on a different scale. The Y-axis can be specified not from "0", but from "1000" to "1300". This will push the lines of the stretch charts apart. In the discussion and conclusions, it is necessary to quantitatively indicate how much the strength decreased and the plasticity increased when the substrate was heated to 600 °С.)

Response: As Reviewer suggested that we have improved the quality of the Figure 8, and added the statement “The ultimate tensile and yield strength decrease by 1.5% and 6.7%, respectively, and the elongation increases by 66.0% when the powder bed is heated to 600 °С.”

  1. Response to comment:( The work does not indicate in which direction, according to the sample growth scheme, samples for mechanical tests were cut. This has a significant effect on the values of the mechanical properties.)

Response: We are very sorry for the negligence of the building direction statements. And we have added the description as shown in Fig.2. We are greatly inspired by the reference 5 in the manuscript and we will study the different mechanical performance in different additive direction in the future work.

  1. Response to comment:( The photo in Figure 4 can be improved.)

Response: As Reviewer suggested that we have improved the Figure 4

  1. Response to comment:(The article has a large number of design flaws that need to be eliminated)

Response: We are very sorry for not explaining the design process clearly. In the present research, it is easy to form martensitic microstructure in Ti-6Al-4V alloy during manufacturing, causing poor plasticity. Therefore, we try to tailor the microstructure by increasing the bed temperature. The formation of β phase and microstructure coarsening occurred with the increasing powder bed temperature. The tensile results show that the ultimate tensile strength and yield strength reduce a little, whereas the ductility is improved dramatically with the increasing temperature when it is higher than 400 ℃. In addition, the abstract has been revised based on this design idea.

In a word, the reviewer’s comments are quite helpful and meaningful, and we have revised our manuscript point-by-point. Thank you for your reviewing my paper and giving your comments again.

Reviewer 2 Report

The manuscript concerns the properties of Ti-6Al-4V alloy produced by the LPBF method. Thus, the article deals with current research problems. The article is interesting. Let me make a few comments:
1. The title is spelled correctly
2. The Abstract provides sufficient information to inform the reader of the assumptions and the results obtained
3. Minor editing errors, e.g. bold text in the Abstact section,
4. The authors cite the publications in the Introduction. They could be better described.
5. "Nearly all the particles are spherical and most of the particles have the diameter of no larger than 53 mm, as shown in Fig.1." You sure that mm?
6. Figure 1. Why is the name SEM Maps? This is not a mapping but a powder morphology.
7. How was the powder layer thickness measured?
8. "The microstructures of the powders (...) were characterized by the scanning electron microscope..." How? Are you sure about the microstructure? How could you study the microstructure of the powder? 
9. "Dog-bone shaped" - why not specify the dimensions of the drawing better?
10. I have big doubts about Figure 3. The marked area is in my opinion the background. You cannot look for peaks on the XRD spectrum that are not there. In my opinion, Figure 3 should be changed.
11. Figure 4 and Figure 7. Please separate the individual microstructures. Everything blends together.
12. Pay attention to the formatting of the text
13. Conclusions are supported by research results.

I would like to see the manuscript after the changes have been made. 

Author Response

Dear Editors and Reviewers:

Thank you for your reviewing and giving your kind and meaningful comments concerning our manuscript entitled “Influence of powder bed temperature on the microstructure and mechanical properties of Ti-6Al-4V alloy fabricated via la-ser powder bed fusion” (ID: materials-1191348). The comments are all valuable and very helpful for revising and improving our paper, as well as the important guiding significance to our researches. We have studied the comments carefully and have made some corrections which we hope meet with approval. Revised section is clearly highlighted using the "Track Changes" function in Microsoft Word. The main corrections in the paper and the responds to the reviewer’s comments are as flowing:

Responds to the reviewer’s comments:

  1. Response to comment:( The authors cite the publications in the Introduction. They could be better described.)

Response: Considering the Reviewer’s suggestion, we have modified the introduction.

  1. Response to comment:( "Nearly all the particles are spherical and most of the particles have the diameter of no larger than 53 mm, as shown in Fig.1." You sure that mm?)

Response: Thanks for your reminding, we have corrected this error to “μm” in this manuscript.

  1. Response to comment:( Figure 1. Why is the name SEM Maps? This is not a mapping but a powder morphology.)

Response: It is true as Reviewer’s comments, and we have corrected “SEM mapping” to “morphology” in the manuscript.

  1. Response to comment:( How was the powder layer thickness measured?)

Response: We are sorry for not explaining the powder layer clearly. Fig.1 showed the morphology of the powder sticked to conductive adhesive, and it’s a very thin layer. In the manuscript, the powder layer thickness is the thickness between two layers during manufacturing, which is controlled by the laser powder bed fusion facility equipped high precision displacement sensor.

  1. Response to comment:( The microstructures of the powders (...) were characterized by the scanning electron microscope..." How? Are you sure about the microstructure? How could you study the microstructure of the powder?)

Response: We are sorry for the incorrect description. And we have corrected it to “The morphology of the powders and the microstructures of the LPBFed samples were characterized by scanning electron microscope”

  1. Response to comment:( Dog-bone shaped" - why not specify the dimensions of the drawing better?)

Response: Considering the Reviewer’s suggestion, we have added the Fig.2.

  1. Response to comment:( I have big doubts about Figure 3. The marked area is in my opinion the background. You cannot look for peaks on the XRD spectrum that are not there. In my opinion, Figure 3 should be changed.)

Response: We are sorry for not explaining the Figure 3 (Now is Figure 4) clearly. The XRD spectrums located in the left and right are obtained by different scanning speed. The XRD spectrums located in the left was obtained between 32 and 65° with a step size of 0.02° and a counting time of 1 s/step, which is a coarse scanning strategy and displays a severe signal-to-noise ratio. The β phase is difficult to be detected in this speed. However, the XRD spectrums located in the right was obtained between 55 and 60° with a step size of 0.01° and a counting time of 5 s/step, which is a fine scanning strategy and displays a slight signal-to-noise ratio. And the β phase could be detected well in this condition. These statements have been added in the manuscript.

  1. Response to comment:( Figure 4 and Figure 7. Please separate the individual microstructures. Everything blends together.)

Response: Thanks for your suggestion. Figure 4 shows the microstructural evolution during LBPF with the increasing powder bed temperature using BSE. Figure 7 shows the quantificational microstructure coarsening with the increasing powder bed temperature, and the average aspect ratios of laths are listed in Table 3, which confirms the microstructure coarsening in Figure 7.

  1. Response to comment: (Pay attention to the formatting of the text)

Response: Thanks for your suggestion, the editor have made formatting changes in my manuscript.

In a word, the reviewer’s comments are quite helpful and meaningful, and we have revised our manuscript point-by-point. Thank you for your reviewing my paper and giving your comments again.

Reviewer 3 Report

The current study investigates the effect of temperature of the powder bed on the resulting mechanical properties and microstructure of additively manufactured titanium alloy. The authors claim to have achieved density of 99.4% when the temperature of the powder bed is below 400 degrees. The authors discuss the microstructure of the fabricated samples.

Please avoid using words such as unfortunately in the abstract and elsewhere in the manuscript. Instead say, having martensitic phase is undesirable in fabricated metallic alloys or something similar. Try to use scientific  wording

Please try to make the abstract concise and to the point

Please consider reviewing the abstract and highlight the novelty, major findings and conclusions.

The authors must rephrase and update their abstract, at its current state it does not read well

Please remove this sentence “In a word, it can be concluded that the powder bed temperature significantly influences the microstructure and mechanical properties of the Ti-6Al-4V alloy manufactured by LPBF, which is an effective method to tailor the LPBFed Ti6Al-4V alloy microstructure to obtain an excellent combination of mechanical properties for various applications” it does not add any value to it at all.

Please avoid using unnecessary wording such as “In a word,” this does not add any value or meaningful value and just makes the sentences a bit longer.

Literature review is limited and needs to be further explored and elaborated, please discuss about past studies similar to your work, discuss what they have done and what were their main findings then highlight how does your current work brings new knowledge and difference to the field and industry.

What is the research gap did you find from the previous researchers in your field? Mention it properly. It will improve the strength of the article.

Figure 1 did you by any chance measure the particle size distribution by a particle sizer? It is more meaningful giving such details instead of an SEM image of powder particles.

Table 1 where did the authors get this data from? If measured by yourselves its fine but if not then please reference it

How many samples were fabricated? How many time were each sample replicated, did the authors use fresh powder each time or reused the used powder? What is the build parameters and standards used (please add them in a table for clarity, I see you mentioned them in the text), there are so many missing details and information in this study.

What was the reason behind those build parameters, are they recommended or chosen based on the machine limitations or something else?

Please add some images and figures showing the experimental setup and equipment used, fabricated samples and post processing of them (i.e. such as polishing or surface processing..etc) this is an experimental study and it needs to clearly describe what has been done in this work.

In section “Densification ratio” you discuss what we see in the graph but there is not critical discussion at all or comparing your current results against past studies which did similar work on titanium or other metals? Can you please elaborate further, also do this in other sections, not just highlight the obersvation we can clearly see from the figures

Figure 4 please separate the figures by yellow or red borders it was a bit confusing to check each image separately

“To further identify the characteristics of α/α´ laths in statistics, the EBSD maps of Ti-6Al-4V alloy after LPBF at different powder bed temperatures are shown in Fig.” why you right this in bold? Is it special from the other text?

Figure 7 please add some arrows and text to tell the readers what are we looking at here in these images

I would strongly recommend you combine the discussion with the results it is standing there separately at the end (suggestion)

Author Response

Dear Editors and Reviewers:

Thank you for your reviewing and giving your kind and meaningful comments concerning our manuscript entitled “Influence of powder bed temperature on the microstructure and mechanical properties of Ti-6Al-4V alloy fabricated via la-ser powder bed fusion” (ID: materials-1191348). The comments are all valuable and very helpful for revising and improving our paper, as well as the important guiding significance to our researches. We have studied the comments carefully and have made some corrections which we hope meet with approval. Revised section is clearly highlighted using the "Track Changes" function in Microsoft Word. The main corrections in the paper and the responds to the reviewer’s comments are as flowing:

Responds to the reviewer’s comments:

  1. Response to comment:( Please avoid using words such as unfortunately in the abstract and elsewhere in the manuscript. Instead say, having martensitic phase is undesirable in fabricated metallic alloys or something similar. Try to use scientific wording)

Response: Thanks for your suggestion, we have revised the related problems.

  1. Response to comment:( Please try to make the abstract concise and to the point, consider reviewing the abstract and highlight the novelty, major findings and conclusions)

Response: We are so sorry for the unclear abstract. In the present research, it is easy to form martensitic microstructure in Ti-6Al-4V alloy during manufacturing, causing poor plasticity. Therefore, we try to tailor the microstructure by increasing the bed temperature. The formation of β phase and microstructure coarsening occurred with the increasing powder bed temperature. The tensile results show that the ultimate tensile strength and yield strength reduce a little, whereas the ductility is improved dramatically with the increasing temperature when it is higher than 400 ℃. And the abstract has been revised.

  1. Response to comment:( Please remove this sentence “In a word, it can be concluded that the powder bed temperature significantly influences the microstructure and mechanical properties of the Ti-6Al-4V alloy manufactured by LPBF, which is an effective method to tailor the LPBFed Ti6Al-4V alloy microstructure to obtain an excellent combination of mechanical properties for various applications” it does not add any value to it at all.)

Response: Considering the Reviewer’s suggestion, we have deleted the word.

  1. Response to comment:( Literature review is limited and needs to be further explored and elaborated, please discuss about past studies similar to your work, discuss what they have done and what were their main findings then highlight how does your current work brings new knowledge and difference to the field and industry.)

Response: Considering the Reviewer’s suggestion, we have modified the introduction. Most of the conventional available LPBF systems can only heat the powder bed to less than 200 ℃, at which, the residual stress forming during LPBF can be reduced. However, this low temperature hardly induces the α' martensite to decompose into α and β phase.

  1. Bruckner et al proposed a model to describe the influence of pre-heating powder bed temperature on the thermometallurgical phenomena in laser cladding initially. B. Vrancken et al experimentally studied the effect of powder bed temperature on the microstructure evolution of Ti-6Al-4V alloy, and found that there is little deposition of α′ martensite when the bed temperature increased to 400 ℃. This is the first time to report the α′ martensite decomposition of Ti-6Al-4V alloy during LPBF. In 2017, Haider Ai et al investigated the martensitic decomposition and mechanical properties of Ti-6Al-4V alloy using the powder bed with the temperature of up to 770 ℃ during LPBF. They found the α′ martensite could decompose into α+β phase when the bed temperature increased to 570 ℃. However, there exist some cavities in the LPBFed Ti-6Al-4V alloy sample, which might damage the mechanical properties. In addition, it is also limited in the systematic studying on the microstructural evolution (β phase characteristic) with the increasing the powder bed temperature during LPBF, and the relationship between the microstructure and mechanical properties should also be further revealed.
  2. Response to comment:( What is the research gap did you find from the previous researchers in your field? Mention it properly. It will improve the strength of the article.)

Response: Thanks for your suggestion, we have added these contents in the introduction.

  1. Response to comment:( Figure 1 did you by any chance measure the particle size distribution by a particle sizer? It is more meaningful giving such details instead of an SEM image of powder particles.)

Response: Thanks for your suggestion, to present the powder size, we tested the powder size distribution by Mastersizer 2000 laser particle size analyzer, as shown in Fig.1(c).

  1. Response to comment:( Table 1 where did the authors get this data from? If measured by yourselves its fine but if not then please reference it)

Response: We are sorry for no explaining the Table 1 clearly. The result was obtained by a X-ray fluorescence spectrometer (XRF-1500). We have added the statement to the manuscript.

  1. Response to comment:( How many samples were fabricated? How many time were each sample replicated, did the authors use fresh powder each time or reused the used powder? What is the build parameters and standards used (please add them in a table for clarity, I see you mentioned them in the text), there are so many missing details and information in this study.)

Response: Thanks for your suggestion, we have added the missing details in the manuscript.

  1. Response to comment: (What was the reason behind those build parameters, are they recommended or chosen based on the machine limitations or something else?)

Response: Thanks for your suggestion. In fact, we have also studied the effect of different energy densities on density and microstructure. However, a large number of experiments are lacked to do a systematic research. And the study would be continued detailedly in our future work. We found the densification ratio is high in the present building parameters. And the powder bed can only be heated up to 600 ℃ in the machine used in the present study, it has been presented in Experimental material and Procedures。

  1. Response to comment: (Please add some images and figures showing the experimental setup and equipment used, fabricated samples and post processing of them (i.e. such as polishing or surface processing..etc) this is an experimental study and it needs to clearly describe what has been done in this work.)

Response: Thanks for your suggestion, the experimental equipment have been added in the manuscript, and the figures are shown here. We think it’s are all common equipment in material research, and it may be not necessary to present in the manuscript. But this is just our opinion, if you have different idea, please don’t hesitate to tell us.

Figure 1. The experimental equipment (a) fabricated equipment (SLM 280 HL facility) (b) Mechanical polishing equipment (PlanarMet300) (c) Scanning electron microscope (TESCAN) (d) Ion polishing system (Gatan-made Precision)

In a word, the reviewer’s comments are quite helpful and meaningful, and we have revised our manuscript point-by-point. Thank you for your reviewing my paper and giving your comments again.

Round 2

Reviewer 2 Report

Many thanks to the authors for taking into account my comments. Thank you also for clarifying all things that were not clear to me. The article is now ready for publication in my opinion. 

Author Response

Thank you for your reviewing my paper and giving your comments again!

Reviewer 3 Report

I am unable to clearly see the changes with the authors showing track changes, please highlight all changes done using yellow color and remove all the track changes in the manuscript, it is very confusing and difficult to follow.

Author Response

We are sorry for the confusing changes, and the track changes were the request of the magazine. But we have removed all the track changes and highlighted all changes done using yellow color in the latest version.

Round 3

Reviewer 3 Report

All questions answered paper can be accepted for publication